

# Brief communication: Mountain permafrost acts as an aquiclude during an infiltration experiment monitored with ERT time-lapse measurements

Mirko Pavoni[1], Jacopo Boaga[1], Alberto Carrera[2], Giulia Zuecco[3], Luca Carturan[3] and Matteo Zumiani[4]

[1] Department of Geosciences, University of Padova, Padua, Italy
[2] Department of Agronomy, Food, Natural Resources, Animals and Environment, University of Padova, Legnaro (PD), Italy
[3] Department of Land, Environment, Agriculture and Forestry, University of Padova, Legnaro (PD), Italy
[4] Servizio Geologico, Provincia Autonoma di Trento, Italy

*Correspondence to*: Mirko Pavoni (mirko.pavoni@phd.unipd.it)

**Abstract.** Continuous frozen layers within the subsoil are generally assumed to act as aquicludes or aquitards. So far, this behavior has been mainly defined analyzing the geochemical characteristics of spring waters. In this work, for the first time, we experimentally confirmed this assumption by executing an infiltration test in a rock glacier of the Southern Alps, Italy. Time-lapse electrical tomography (ERT) technique was adopted to monitor the infiltration of a huge amount of water spilled on the surface of the rock glacier. 24 hours ERT monitoring highlighted that the injected water was not able to infiltrate into the underlying frozen layer.

## 1 Introduction

In alpine regions, groundwater originating from moraines and rock glaciers is highly contributing to the streamflow (Wagner et al. 2016). Therefore, a key factor in the hydrological modeling of alpine catchments is the determination of the hydraulic properties of these landforms. The subsoil hydrodynamic of moraines, talus and hillslope aquifers is relatively well known. On the other hand, the hydraulic behavior of rock glaciers and their impact on the hydrology of alpine catchments are relatively less defined (Pauritsch et al., 2017). The hydrological and the geochemical monitoring of spring waters emerging downslope of active rock glaciers have been used to investigate runoff processes and the presence and role of frozen layers in alpine catchments (e.g., Krainer et al., 2007; Carturan et al., 2016; Brighenti et al., 2021). In active or ice-rich intact rock glaciers, continuous frozen layers are typically considered as aquicludes (Giardino et al. 1992). Krainer et al. (2007) separated a subsurface flow component, derived from snow-ice melting and rainwater, and a deeper and longer stored aquifer at the bottom of Reichenkar active rock glacier (Austrian Alps). Harrington et al. (2018) defined the inactive Helen Creek rock glacier (Alberta, Canada) as an unconfined aquifer, as the limited ground ice distribution is unlikely to act as pure aquiclude. These investigations suggest that rock glaciers host complex and heterogeneous aquifers with a layered internal structure. Nevertheless, geochemical surveys have not the ability to accurately define the acquirer's model (e.g., layers thickness, discontinuities, and lateral/vertical heterogeneities) if not integrated with geophysical surveys.

To verify and confirm the hydraulic behavior of the frozen layer, an infiltration experiment combined with electrical resistivity tomography (ERT) time-lapse measurements has been performed in the inactive Sadole rock glacier (Southern Alps, Italy). 800 liters of salt water have been injected at the surface of the rock glacier and the results of the ERT time-lapse monitoring confirmed that the flow of the injected water did not cross the frozen layer existing beneath. Controlled irrigation experiments combined with ERT time-lapse measurements were successfully applied to study vadose zones (Cassiani et al., 2006), or the more challenging hillslope catchment (Cassiani et al., 2009). The Sadole rock glacier infiltration experiment represents the first attempt to adopt this monitoring technique to the mountain permafrost environment and, considering the promising results,





can be used as reference to improve future tests in intact rock glaciers to better characterize and quantify the hydraulic
properties of the frozen subsoils.

## 2 Site description

The Sadole rock glacier is located in the Sadole Valley, a lateral of the Fiemme Valley, in the Eastern part of the Trento
Province (North-East Italy, Fig.1A). The rock glacier ranges between 1820 m a.s.l. and 2090 m a.s.l. and feeds the spring of
the Rio Sadole, a tributary of the Avisio River in the Adige River catchment. The Sadole rock glacier is a complex periglacial
landform that derives from the confluence of three different rock glaciers (Fig.1B). The main rock glacier body, whose front
reaches the minimum elevation of 1820 m a.s.l., is partly overridden by two smaller (and younger) rock glaciers at the
orographic left and right sides. This periglacial landform occupies the floor of two coalescent glacial cirques. Steep rock walls
and sharp crests almost entirely bound these cirques, with the exception of the Sadole Pass that was likely a glacial transfluence
saddle during the last glaciation. Slope deposits are found between the rock walls and the rock glacier rooting areas. These
deposits have gravitational or mixed gravitational/debris-flow/avalanche origin and are predominantly active. From a
geological point of view, the rock glacier is composed of magmatic rocks (riodacitic ignimbrites) that belong to the Athesian
Volcanic Group, a late-Paleozoic (Permian) volcanic succession. The Sadole rock glacier has been classified as 'relict' in the
inventory of Trento Province (Seppi et al., 2012), due to the low altitude and to the vegetation cover (Fig.1C). Despite this,
the general convex morphology and the low water temperature of its spring, ranging between 1.0 and 1.8°C in the ablation
season (May to October), suggested that it may preserve ice and/or permafrost inside (Carturan et al., 2016 and references
therein). The preservation of internal frozen layers is ensured by the thick mantle of boulders that cover its surface (Morard et
al., 2008). In addition, ice outcrops have been observed in mid-summer two meters below the surface, in a pit dug during the
1$^{st}$ World War (green dot in Fig.1B). Geophysical surveys have been performed in summer 2021 on the main rock glacier
body. Several ERT transects (brown lines in Fig.1B) have been collected, confirming the presence of a discontinuous frozen
layer at a depth of about 10 meters. Consequently, soil temperature sensors (red dots in Fig.1B) have been installed in different
location of the rock glacier bodies and an intensive monitoring system of the spring water (geochemical and temperature) has
been set. To evaluate the hydraulic behavior of the frozen layer, an infiltration experiment with ERT time-lapse measurements
has been realized in June 2022. The ERT monitoring transect (orange line in Fig.1C) has been located in the same area of the
2021 ERT surveys and its orientation has been chosen considering the maximum slope gradient.

## 3 Methods

### 3.1 Experiment principles

ERT surveys are performed to detect the electrical properties of the ground. The method can be used for monitoring time-
dependent subsurface processes by repeating periodically the measurements using the same electrode array (Binley, 2015).
This ERT data acquisition method is defined as "ERT time-lapse technique", and can be performed with controlled irrigation
experiments (Cassiani et al., 2006; Cassiani et al., 2009). In these tests, a large amount of salt water (usually several hundred
liters) is released into the subsoil system, and the propagation of the injected water is investigated using the ERT time lapse
survey. A dataset of apparent resistivities is collected before the injection, at a time called time zero (t0). Subsequently, as the
salt water propagates into the ground, new ERT datasets are periodically acquired at defined time steps (t1, t2, …, tn). The
changes of electrical properties in the subsoil, due to the injected water flow, are usually not clearly highlighted by comparing
the single measured apparent resistivities of the collected datasets, or by comparing the individual inverted resistivity models.
To enhance the variation from one-time step to the next, only the inverted model t0 is represented in terms of absolute
resistivities, while the other time steps results are plotted in terms of percentage variations of resistivity with respect to the t0
initial model (Binley, 2015).

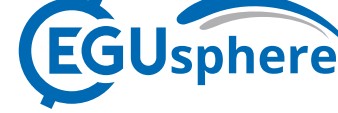

### 3.2 Data acquisition

The infiltration experiment on the Sadole rock glacier was carried out in June 2022. The ERT monitoring transect (orange line in Fig.1C) has been located in the same area of the previous 2021 ERT surveys, and its orientation has been chosen considering the maximum slope gradient. We set an ERT survey line of 72 electrodes, spaced 1.5 m from each other, for a total array length of 106.5 m. The measurements have been performed with a Syscal Pro georesistivimeter (Iris Instruments), using a dipole-dipole configuration with different skips (1, 3, 5 and 7 - the skip represents the number of electrodes skipped to create a dipole), and a stacking range between 3 and 6 with 5% error threshold. The chosen configuration allowed to collect direct and reciprocals measurements (by exchanging current and potentiometric dipoles) and to estimate a reliable experimental error for the acquired datasets (Binley, 2015). The error is usually high in rock glacier environments, due to the high contact resistances between electrodes and boulders (Hauck & Kneisell, 2008). To partially overcome this problem, and increase the amount of injected current (Pavoni et al., 2022), we inserted the electrodes between the boulders using sponges soaked with saltwater (Fig.2A). The sponges have been wetted at the beginning of each measurement during the ERT time lapse survey, to reach (approximately) homogeneous contact resistances for each collected dataset. Collecting measurements with different contact resistances could lead in fact to changes in resistivity models not linked to the flow of the injected water. The water for the experiment has been collected during the previous months, using ten 100-liter bins. The bins have been placed on the Sadole rock glacier, in the point selected for the water injection, in the early spring, when snow cover was still present (Fig.2B). They were filled with snow and covered with nylon sheets pierced at their center to collect rainwater. This way, in mid-June the bins were completely filled with a mixture of snowmelt and rainwater. Before the experiment 3 kg of NaCl were added to each bin to obtain a salt water solution. The water injection point has been chosen considering the results of the 2021 ERT surveys (brown lines Fig.1B), which enabled to detect high resistivities (>50 kΩm) at a depth of about 10 meters, suggesting the presence of a frozen layer. The new survey line was oriented along to the maximum slope and centered with respect to the position of the bins (Fig.1C), ensuring the maximum penetration depth (about 20 meters) below the injection point. After collecting the t0 dataset, 8 bins were emptied one after the other, injecting 800 liters of salt water into the subsoil system (Fig.2C). The remaining water has been used to wet the sponges before each new data acquisition. Four datasets have been acquired in the first hour, followed by four datasets at hourly intervals, and a last dataset collected 24 hours after water injection. No rain or uncontrolled water contribution happened during the experiment.

### 3.3 Data processing

All the acquired datasets have been filtered removing quadrupoles with standard deviation higher than 5% and quadrupoles with measured apparent resistivity higher than 100 kΩ*m. Only the common quadrupoles saved in all the filtered datasets have been used to perform the inversion process of each dataset. The inversion of each dataset has been performed using the Python-based software ResIPy (Blanchy et al., 2020), based on Occam's inversion method (Binley, 2015). The code minimizes an objective function that quantifies the misfit between the measured dataset and the predictions made by the electrical resistivity subsoil model. An expected data error of 20% for the inversion processes has been evaluated after the reciprocal check (Binley, 2015). Once a common unstructured triangular mesh has been created, all the acquired datasets have been independently inverted. Only the t0 initial model was plotted in terms of absolute electrical resistivity, while the other models obtained with the ERT time-lapse survey were plotted as percentage variations in resistivity compared to the initial model t0. Note that, slight changes in the order of 10% have low reliability, since they can be linked to different factors (e.g., inversion artefacts, instrumental error and non-perfectly homogeneous contact resistances during different time of acquisition) and not necessarily to the flow of the injected water. To detect the frozen layer boundary in t0 (i.e., the thickness of the active layer), we applied the steepest gradient method (Chambers, 2012), calculating the second derivatives point between the lowest and greatest values of resistivity in the vertical direction. This method, as suggested by forward modeling, is the most reliable to evaluate the thickness of the active layer (Herring et al., 2022).



### 4 Results

Figure 3A shows the resistivity section at t0. The high resistivities ($\rho$>30 k$\Omega$*m) close to the surface are linked to the voids among coarse debris and blocks (Fig.2A), typical in rock glacier environments (Hauck and Kneisel, 2008). Below the top high resistivity layer, lower values of resistivity ($\rho$<10 k$\Omega$*m) are found and can be associated with a decrease in porosity and grain size of the deposit, and a possible increase in humidity. At the South-West and North-East edges of the section this low resistivity layer reaches the bottom of the model. On the other hand, in the central part of the model (30<x<70 m) a clear change is detected at a depth of about 10 meters. Below this boundary, the resistivity rapidly increases ($\rho$>50 k$\Omega$*m), highlighting the presence of a frozen layer (Hauck and Kneisell, 2008). By applying the gradient method in the vertical direction, we defined 55 k$\Omega$*m as the boundary of the permafrost layer and the same value has been used to define its lateral termination. Note that, in Fig.3 the assumed frozen layer boundary has been highlighted with a black dashed line.

In Fig. 3B high negative resistivity variations (-100 %< $\Delta\rho$ < -80 %) show a quick infiltration of the injected water up to a depth of 10 meters within the first 15 minutes after the injection. This wet area persisted below the injection point until the last survey (t10 – Fig.3M), even if it seems to slowly shrink from one data acquisition to the next. Resistivity variations between -10 and -40 % in Fig.3B and Fig.3C indicate a vertical infiltration of the injected water in the area upslope the permafrost body (x<30 m), but also a lateral downslope subsurface flow (in the north-east direction) above the identified frozen layer. Where the frozen layer ends (x $\approx$ 70 m), the water clearly appears to be able to propagate deeper (Fig.3D-3M). Concerning the upslope area (x<30 m), the negative resistivity variations are found from the surface to the bottom of the section until t4 (see Fig.3B-3F), highlighting a main initial vertical infiltration of the injected water. In the next time steps the negative values develop mainly at few meters of depth (see Fig.3F-3L), showing a possible anomalous lateral subsurface flow (south-west direction). This lateral negative variation upslope is still present 24 hours after the injection (t10 – Fig. 3M) and, at the same time, the water is still flowing downslope in north-east direction.

Inside the defined frozen layer, the negative resistivity variations are practically null, only at its edges few negative resistivity variations are found for t1, t2, t3, t4 and t10. Consequently, it seems that the injected water did not propagate through the frozen layer and mainly surrounded it, keeping the subsurface flow in the north-east direction. Marked local resistivity variations (between -70 and -100 %) were found for t7 and t8 (Fig.3H and Fig.3I), in the layer with lower initial resistivities ($\rho$<10 k$\Omega$*m – Fig.3A) above the frozen layer.

Note that, the salt water injected during the experiment did not affect the electrical conductivity of waters at the rock glacier spring, which is monitored at hourly intervals by a datalogger.

### 4 Discussion and conclusions

The resistivity variations detected by the ERT time-lapse surveys were useful to identify: i) the infiltration of the injected water in the subsoil, ii) the formation of subsurface flow in the surroundings of the frozen layer, and iii) the main direction of subsurface flow towards north-east. The resistivity variations (from -80 to -100 %) observed for t1 close to the injection area, up to a depth of about 10 meters, indicate a rapid vertical infiltration of the water due to the presence of boulders, fractures and coarse sediments with high vertical permeability. The large amount of water rapidly injected has probably saturated this area, which has become the source of the subsurface flow. Although we do not have any measurements of saturated hydraulic conductivity, we can speculate that hydraulic conductivities may be much higher (in the order of $10^{-2}$ m/s) than the ones observed in shallow soil layers of young moraines (consisting of coarse and fine sediments) as found in the Swiss Alps (Maier et al., 2021). Subsurface flow, moving downslope along the north-east direction, is likely originated at the boundaries between large boulders and a finer sediment layer (Mandal et al., 2005). This layer is in fact characterized by lower resistivities in t0 ($\rho$<10 k$\Omega$*m - Fig.3A) compared to the shallower depths, and has likely lower permeability. Furthermore, the presence of boulders and large rocks at various depths can lead to local reductions of the permeable area, causing funnel flow, and/or a



splitting of flow paths (Hartmann et al., 2020). Splitting flow paths, due to the presence of large boulders, may have determined
the infiltration of some injected water along the south-west direction and a local accumulation upslope of the frozen layer,
which resulted in the observed negative resistivity variations. From t1 to t5 the resistivity variations suggest almost a
continuous subsurface flow in the north-east direction along the maximum slope gradient, whereas from t6 to t10 local negative
resistivity variations indicate the accumulation of injected water in areas where there is a likely local change in permeability.
In these areas, the injected water may reside for a longer time compared to the other subsurface zones having higher hydraulic
conductivities.
In the last decades, the hydraulic behavior of frozen layers has been defined mainly by studying the response of the rock
glaciers springs to geochemical tracers (e.g., stable isotopes of hydrogen and oxygen, electrical conductivity and radionuclides
– e.g., Krainer et al., 2007; Brighenti et al., 2021). Therefore, the most interesting result of this infiltration experiment with
ERT time-lapse measurements is that negative resistivity variations, related to the injected water flow, are almost negligible
inside the detected frozen layer. This confirms the low permeability and the assumption that a continuous permafrost layer can
act as an aquiclude or aquitard (Giardino et al. 1992; Harrington et al., 2018). Furthermore, the monitored flow of the injected
water confirms high lateral and vertical heterogeneities in mountain permafrost subsoils, as revealed by continuous drillings
by Krainer et al. (2012).
Future development of the current work is to perform a similar experiment on an active rock glacier, with a continuous frozen
layer, and by injecting the salt water solution using different intensities to simulate a variety of rainfall events. Moreover, the
acquisition times of the ERT measurements could also be modified, i.e. collecting datasets for a longer period until the subsoil
system returns completely to the pre-injection conditions, even if this can be complicated by possible rainfall contribution
during the experiment. This way a better evaluation and quantification of the hydraulic conductivity in the active layer of rock
glaciers could be achieved.

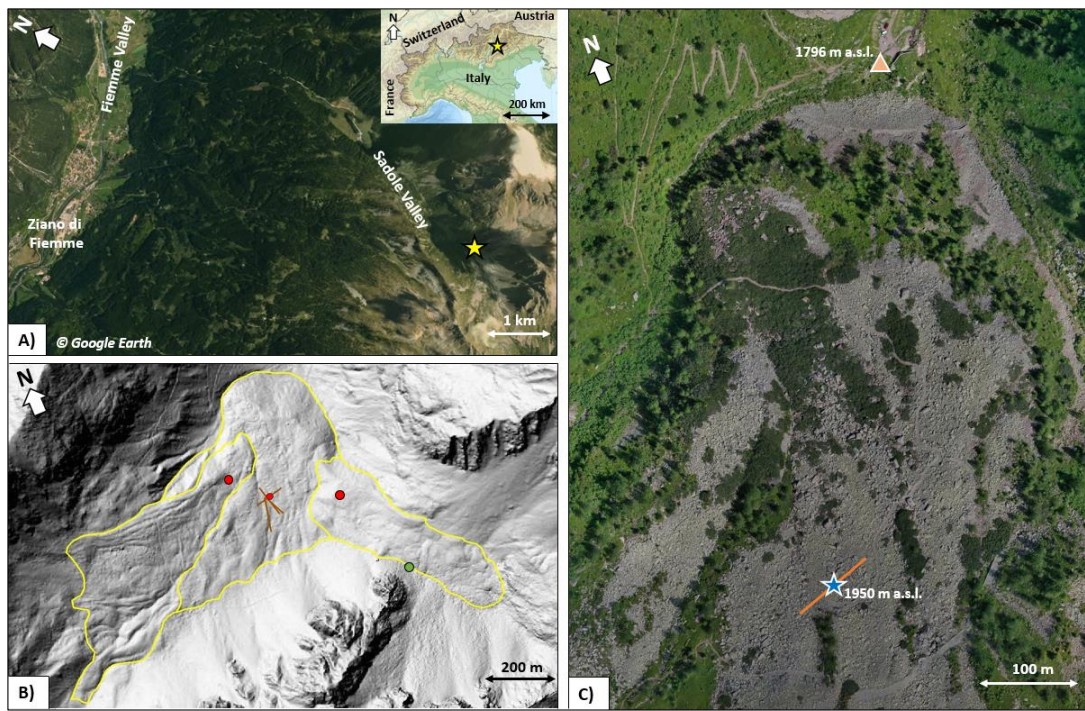


**Figure 1: A) Geographic location of the Sadole rock glacier (yellow star), adapted from © Google Earth Pro and Italian Physical**
**Map produced by The University of Texas at Austin; B) hillshaded LiDAR DEM (modified from WebGIS PAT – Provincia**
**Autonoma di Trento) showing the three different units that compose the Sadole rock glacier (yellow lines). Brown lines represent**





ERT surveys performed in summer 2021, red circles defines the position of soil temperature sensors, and green circle is the location
of the   Austrian well (1st World War); C) Orthophoto (Commissione Glaciologica SAT, 2022) showing the ERT transect (orange
line) used for the infiltration experiment, the salt water injection point (blue star – 1950 m a.s.l.), and the location of the rock glacier
spring (brownish triangle – 1796 m a.s.l.).

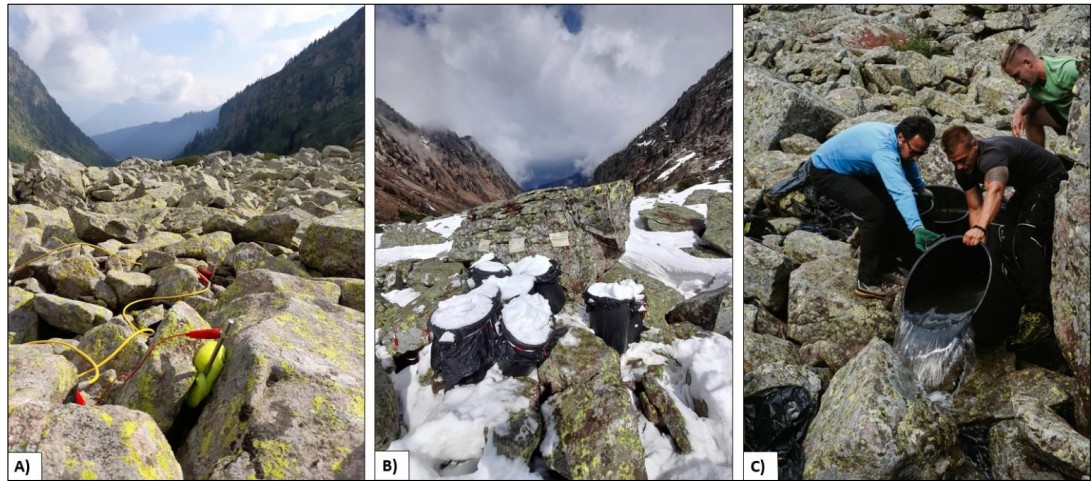


Figure 2: A) Electrodes inserted between the boulders using sponges soaked with saltwater to improve the contact resistances of the
ERT surveys; B) 10 bins placed at the selected injection point in early spring 2022, filled with snow and covered with nylon sheets
pierced at their center to collect rainwater; C) Injection of 800 liters of salt water into the subsoil system.

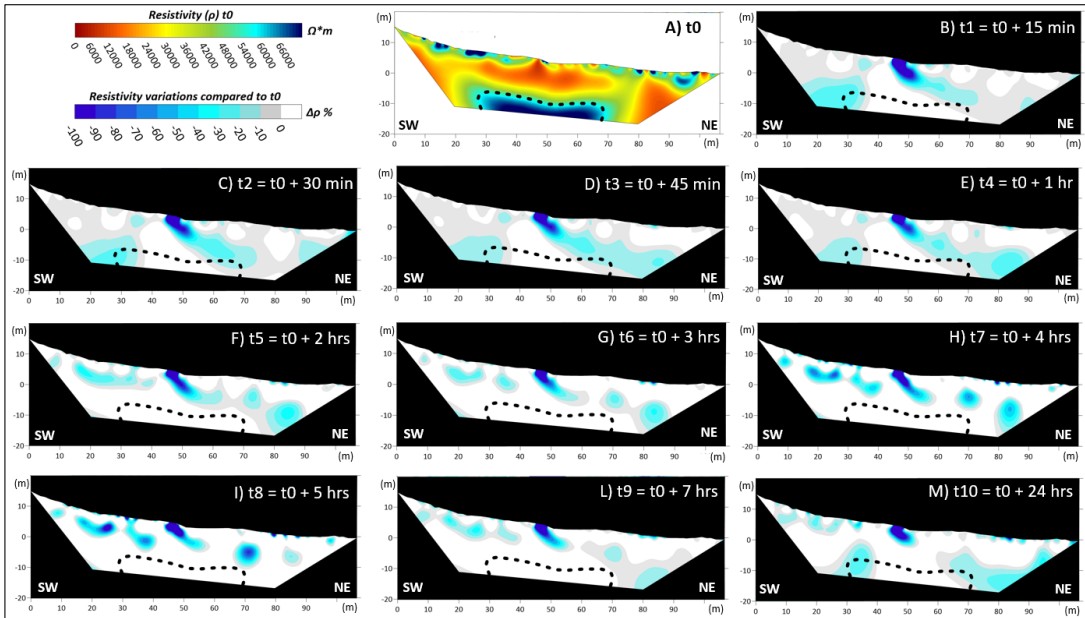


Figure 3: A) Inverted resistivity section calculated from the t0 dataset. Inverted resistivity variations (%), compared to the t0 model
(A), calculated from B) t1 dataset (t0 + 15 minutes), C) t2 dataset (t0 + 30 minutes), D) t3 dataset (t0 + 45 minutes), E) t4 dataset (t0
+ 1 hour), F) t5 dataset (t0 + 2 hours), G) t6 dataset (t0 + 3 hours), H) t7 dataset (t0 + 4 hours), I) t8 dataset (t0 + 5 hours), L) t9
dataset (t0 + 7 hours), and M) t10 dataset (t0 +24 hours). The black dashed line represents the boundary of the frozen layer defined
applying the steepest gradient method to the inverted resistivity model t0.
*Author contributing.* MP, JB, AC and MZ have been involved in data acquisition; MP performed the data processing; LC
realized the geological framework; GZ carried out the interpretation of the results; all authors contributed to the writing and
editing of the manuscript.





*Acknowledgements.* Authors thanks Tommaso and Barbara, managers of "Il rifugio Baita Monte Cauriol", for logistical support; the "Magnifica Comunità di Val di Fiemme" and "Comune di Ziano" for authorizing the investigations; and the "Servizio Geologico della Provincia Autonoma di Trento" for the support.

*Data Availability Statement.* The datasets used to obtain the results presented in this work are avaible at the open source repository: https://zenodo.org/badge/latestdoi/541527187 (DOI: 10.5281/zenodo.7113054).

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
