# Peer review of "Brief communication: Mountain permafrost acts as an aquiclude during an infiltration experiment monitored with ERT time-lapse measurements"

_EGUsphere, 2022_

## Author Response (AR2)

Dear Editor, many thanks for your comments. We modified the manuscript according to your instruction and the text was checked by a mother tongue colleague.

Many thanks to the anonymous Reviewer 1 for the useful and constructive comments that will help us to improve our manuscript.

**Reply point by point to Reviewer 1**

- **General Comments**: the authors have conducted a novel infiltration experiment in a rock glacier that was monitored using a time-lapse ERT method. Results indicated that the underlying frozen layer was impermeable to the infiltrated water, which confirmed previous assumptions only determined from geochemical analysis in spring waters downslope. A weakness of the manuscript is the lack of quantitative analysis beyond the subsurface imagery and the lack of ground truthing data to support the geophysical interpretation. Overall, the manuscript was well written and presented and clearly defined objectives and results. I found the experimental method and application interesting and applicable to a wide range of cryo-hydrogeologic settings and of interest to the wider scientific community.

**Reply**: We thank the anonymous Reviewer for the comment. Quantitative analysis regarding hydraulic conductivity just via time-lapse ERT is very challenging. First of all, as we pointed out in the manuscript, it is well known that mountain permafrost subsoil is very heterogeneous both vertically and laterally, and this is clear also from our results (see discussion and conclusions). This makes very hard to assess representative hydraulic properties for the entire domain. Furthermore, due to logistical problems (adverse weather, very common in those high mountain environments), we could not extend for more time the experiment to when the subsoil conditions became similar to the pre-injection conditions. With that information, it would have been possible to define the time the injected salt water takes to leave the monitored subsoil area, and consequently to define an average hydraulic conductivity in the active layer where the flow mainly occurred. See lines 147-160 in the Revised Manuscript.

- **Specific Comments**: my main criticism is the lack of quantitative analysis from the experiment. From some assumptions of hydraulic gradient based on slope and a range of porosities, could an approximate hydraulic conductivity be derived from the results? Alternatively, could you report transit times/velocities of the leading edge of the salt plume?

**Reply**: The experiment has been performed to verify the low permeability of a permafrost layer and to demonstrate that this layer acts as an aquiclude/aquitard, for the first time with a geophysical method in rock glaciers environments (a very challenging system for this kind of experiment). The aim of our brief communication is to show to the permafrost community that ERT time lapse surveys, historically performed in other and easier environments, can be also applied with success in a rock glacier site. More advanced analyses regarding the hydraulic conductivity of the layers that compose the frozen subsoil are more suited for a complete paper, with no limitations regarding the text and images/tables (as we have in a brief communication). Nevertheless, in the next future, our goal is to further investigate these datasets, perform new ERT infiltration experiments integrated with chemical and dye tracer experiments, and in this way to define a suitable strategy to estimate the hydraulic conductivities in these environments. See lines 147-160 in the Revised Manuscript.

- **Specific Comments**: Another point of note is that while the infiltrating water did not appear to infiltrate through the frozen layer, this layer was not continuous. Is it expected that in active rock glaciers, this would be the case? If this layer is not continuous, its permeability is less important in the context of deep infiltration and recharge.

**Reply**: The anonymous reviewer is right. As we highlighted in the site description chapter, the Sadole rock glacier is not active but inactive. From several ERT surveys performed in summer 2021, we defined a discontinuous frozen layer at a depth of about 10 meters in the area where we performed the infiltration experiment in summer 2022. The survey line for the infiltration experiment has been settled specifically to detect how the injected water flows in the area where the frozen layer is present and how it flows where the frozen layer ends. This allowed us to define a main downstream subsurface flow in the area where the frozen layer is present, while a predominantly vertical infiltration exists where the frozen layer ends. Future studies should aim to perform an ERT time lapse infiltration experiment in an active rock glacier with a continuous frozen layer, where a different infiltration pattern is expected to occur, as suggested by the reviewer. See in the Revised manuscript lines 56-58, 77-80, 157-160, and Fig1 D-E the discontinuos structure of the frozen layer .

- **Specific Comments**: line 14, Suggest adding the 800L instead of "huge amount". Everyone's interpretation of huge will be different.

**Reply**: Thank for the comment, we will do it. See line 14 in Revised Manuscript.

- **Specific Comments**: line 147-148, what timeframe are you referring to? No change during the experiment or following longer-term monitoring afterwards? The distance between the injection point and spring appears to be several hundred meters based on Figure 1c. I am doubtful there would be a detectable change in conductivity by the time the tracer reached the spring, regardless of the low permeability frozen layer

**Reply:** The spring is a source of drinking water used by the local community, therefore it is continuously monitored by the competent authorities. As you correctly stated, no changes in water salinity of the spring were visible during the weeks following our experiment.

- **Specific Comments:** figure 3, Suggest adding the injection location to the plot(s).

**Reply:** Yes, very good suggestion, we will add the injection point in the plots. See new Figure 3 in the revised Manuscript.

- **Specific Comments**: line 30, "acquirer's" should be "aquifer's"

**Reply:** Yes, we will correct it. See line 30 in the Revised Manuscript.

Many thanks to the anonymous Reviewer 2 for the useful and constructive comments that will surely help us to improve the quality of our manuscript.

**Reply point by point to Reviewer 2**

The authors present an interesting approach to increase knowledge about the internal structure and hydraulic processes of rock glaciers. The paper is well written, description of the site, methods and results seem complete, and the results are very promising. I like the original and minimalistic approach to collect and preserve enough water for the experiment.

I have a few concerns regarding potential over-interpretation of the ERT tomogram and the time-lapse results, but in general the paper has an added value and will be interesting to a wider scientific community.

**1) GENERAL COMMENTS**

– How did you calculate the percentage change in resistivity? From absolute resistivity values or from the logarithm of the resistivity values? I strongly recommend using the logarithm for these calculations in order to avoid emphasizing changes in the high resistivity zone:

$$(\log10(res\_t1) - \log10(res\_t0)) / \log10(res\_t0) * 100$$

**Reply:** We really appreciate the suggestion of the anonymous reviewer. In the results presented in Figure 3 of the submitted manuscript, the percentage variations were calculated from absolute resistivity values. We applied the approach proposed by the reviewer and we used the logarithmic resistivity values to define % changes between $t_0$ and other time steps. The new output is very similar to our previous results but, as correctly highlighted by the reviewer, the approach avoids over-emphasizing high variations. Therefore, in the revised version of the manuscript we will present the results of the time-lapse experiment (new Fig.3) obtained applying the approach suggested by the reviewer, which clearly improved the final output of the survey. See Fig. 3 in the Revised Manuscript and lines 104-108.

– Be careful with over-interpretation of zones at the edges and bottom of the tomogram, i.e. in the least sensitive zones. In the context of interpretation of resistivity changes, the publication of Mewes et al. 2017 (https://tc.copernicus.org/articles/11/2957/2017/), might be of interest to you.

**Reply:** We agree with the comment of the reviewer that the bottom and the edges of the tomogram are the least sensitive zones. Nevertheless, we acquired a very high number of measurement points (2594) and even after the applied filtering, the inversion of each dataset was performed with about 2100 measurements, which homogeneously cover the investigated pseudo-section. Furthermore, the acquisition scheme that we applied (dipole-dipole with different skips: 1, 3, 5, 7) has been specifically designed to guarantee high resolution in the shallower part of the subsurface thanks to the lower skips, and a large penetration depth thanks to the higher skips. Considering all this, we think that the negative resistivity variations that we see at the corners of our models, which can be linked to the vertical infiltration of the injected water where the frozen layer is absent, can be reliable. Moreover, we already knew the limits of the frozen layer from the electrical surveys carried out in the summer of 2021 (see Fig.1B). One of these survey lines was parallel to the one used for the infiltration experiment, but it was longer (48 channels with 3m spacing) and it allowed to define vertical and lateral boundaries of the frozen layer along that direction. From this result, it is clear that the frozen layer is discontinuous, and therefore, we defined the position of the water injection and the orientation/length of the survey line, adopted for the infiltration experiment, considering this information. The survey line has been centred with respect to the discontinuous frozen layer and the injection point placed in the middle of the array, where we were sure to find the frozen layer. The expectation was to verify the lateral subsurface flow in the area where the frozen layer develops and vertical infiltration where it is no longer present, as found by the outcome of the experiment. As highlighted in the very interesting paper suggested by the reviewer, which surely will be included in the references of the revised manuscript, it is very challenging, and honestly unrealistic, to perfectly define the flow paths of the injected water by resistivity changes in tomograms. Considering this, in the submitted work we only fixed the limit of the frozen layer using the steepest-gradient method (black dashed line in fig.3). This method, as suggested by recent forward modeling analysis, is the most reliable way to objectively evaluate the thickness of the active layer (Herring et al., 2022). On the other hand, as regards the interpretation of the injected water flow, we did not try in any way to precisely define the water flow paths, but we only highlighted that negative resistivity changes never affect the area where the frozen layer has been defined, suggesting a lateral subsurface flow above the permafrost layer and a vertical infiltration where it is no longer present. Finally, considering the results found in the last time series $t_5$-$t_{10}$, which highlight local areas with high negative % changes, we have interpreted them as areas with finer sediment and lower permeability. In a rock glacier environment, we can expect these "sharp" vertical and lateral granulometric variations (and consequently permeability), as recently highlighted by Phillips et al. (2023) on the Schafberg rock glacier (Engadine, Switzerland) during the coring of two boreholes a few meters apart from each other's.

In the revised manuscript, we think it will be important to add these considerations about the sensitivity and the uncertainties of the results at the edges and bottom of the tomograms, but also to specify that we tried to optimize the survey thanks to the high number of measured points, homogeneously distributed in the pseudo-section, the type of acquisition scheme that we applied (dipole–dipole multi-skip), and the position chosen to perform the infiltration test.

See revised Manuscript lines 56-58, 72-84, 124-131,152-154, and Figs.1 D-E.

**2) SPECIFIC COMMENTS:**

**Introduction:**

- L20: "...relatively well known." --> give references here.

**Reply:** "The subsoil hydrodynamic of moraines, talus and hillslope aquifers is relatively well known. On the other hand, the hydraulic behavior of rock glaciers and their impact on the hydrology of alpine catchments are relatively less defined". Both the sentences in lines 20-22 can be referred to Pauritsch et al. (2017). In this interesting paper, it is possible to find a wider bibliography referring to the hydrological characteristics of storage and transmission of Alpine aquifers. Our manuscript is proposed as a Brief Communication and consequently we have limited number of references that we can use (20), therefore we tried to split them equally in the various chapters of the manuscript. Nevertheless, if requested we can add more references to assert the sentences of lines 20-22.

See lines 20-22 in the Revised Manuscript

- L30: maybe 'aquifer's structure' is better than 'aquifers model'?

**Reply:** We agree with the reviewer, and we will use his/her suggestion in the revised manuscript.

See line 30 in the Revised Manuscript

- L34ff and L37ff contain main results and conclusions, respectively, which is not necessary in the introduction and somehow prevents the reader from reading the rest of the paper. Consider deleting here.

**Reply:** We appreciate the suggestion of the reviewer. We will modify the introduction in order to avoid the information about the final results of the experiment. See new Introduction in the Revised Manuscript without any reference to the results of the survey.

**Site description:**
- can you give a rough estimate of the thickness of the rock glacier in the zone of the ERT profile? This wold be useful to evaluate the depth to bedrock in the tomogram and to check if the resistive layer might be related to base of the rock glacier.

**Reply:** As proposed previously in the second general comments reply, we could add the location of the ERT survey lines performed in summer 2021, and the obtained resistivity section. These survey lines were carried out with 48 electrodes spaced 3 meters, they allowed to define the discontinuos structure of the frozen layer, and partially its vertical and lateral boundaries. Nevertheless, bedrock was not reached in these past surveys. The discontinuos frozen layer develops at an average depth of 10 meters, has a thickness that varies from 5 to 15 meters, and is followed by basal unfrozen till layer. See Fig.1 in the Revised Manuscript.

- L52f: in addition to classification as 'relict' by Seppi et al, it would be good to also refer to the terminology of the recent IPA action group on rock glaciers: (https://bigweb.unifr.ch/Science/Geosciences/Geomorphology/Pub/Website/IPA/Guidelines/V4/220331_Baseline_Concepts_Inventorying_Rock_Glaciers_V4.2.2.pdf)

**Reply:** We thank the reviewer for this suggestion. In the revised manuscript we will add that the rock glacier has neither geomorphological evidence nor detection of current movement associated with permafrost creep. In addition, the development of vegetation (grass, shrubs and mature specimens of Larix decidua) and soil cover, although discontinuous, confirms that the rock glacier can be classified as relict, also in agreement to recent guidelines provided by the IPA Action Group Rock glacier inventories and kinematics (RGIK, 2022). See lines 48-49 in the Revised Manuscript

- L55: better: preserve permafrost ice inside ('and/or' does not make sense here)

**Reply:** We agree and we will modify it in the revised manuscript. See line 50 in the Revised Manuscript

- L59f: can you give a reference for these profiles, or include at least one of them in one of your figures for transparency?

**Reply:** The results of the surveys performed in summer 2021 have not been published, only used in a master thesis. Nevertheless, as proposed previously in the second general comments reply, we could add in the map of the rock glacier, the location of the ERT survey lines performed in summer 2021, and the obtained resistivity section. See Fig.1 in the Revised Manuscript.

**Methods:**

- L77: How did you calculate the percentage resistivity change (see general comment)?

**Reply:** Please see the reply to the first general comment. See new Fig. 3 in the Revised Manuscript and lines 104-108.

- L80: June is very early, the active layer still largely frozen. This may have consequences for the flow paths during the experiment. For future experiments (as mentioned in your outlook) you could consider to conduct them later in the summer (Aug-Oct), in order to test flow paths in real summer conditions, with fully developed active layer.

**Reply:** We totally agree with the comment of the reviewer. Future infiltration experiment will be surely carried out in late summer, particularly if performed in active rock glaciers. Nevertheless, Sadole rock glacier has a particularly large thickness of the upper unfrozen layer (about 10 meters). Moreover, the ERT surveys of 2021 were performed in the middle of September and the results show practically the same subsurface structure as in the ERT survey carried out in the beginning of summer 2022. This is why we do not think that, in this peculiar site, performing the experiment at the end of summer could change drastically the final output of the infiltration survey. Therefore, the result presented in the submitted manuscript can be considered, in our opinion, reliable to define the aquitard behaviour of the discontinuous frozen layer in the Sadole rock glacier. See lines 157-158 in the Revised Manuscript.

- L80ff: 'The ERT monitoring (...) maximum slope gradient." --> repetition of L63f

**Reply:** The reviewer is right and in the revised manuscript we will delete this repetition.

- L101: 'subsoil' is not appropriate here, as there is no soil on a rock glacier. Maybe 'subsurface' is better here and for all later usages of 'subsoil'?

**Reply:** We appreciate the suggestion of the reviewer and we can replace 'subsoil' with 'subsurface' in the revised manuscript. See the Revised Manuscript

- L107: please describe in more detail: Why did you choose the limit of 100 kΩm? How many points did you delete? How was the overall data quality (agreement between normal/reciprocal measurements)?

**Reply:** The limit of 100 kΩm was chosen to delete few high values of $\rho_a$ which systematically occurs in all the acquired datasets in the upper layer of the pseudo-section and related to the large voids between boulders. As we verified, those values are not affecting the final result of the inversion process and the time lapse-survey. Therefore, in the revised manuscript we can avoid mentioning this filtering. On the other hand, the main filtering of the collected datasets is produced by removing quadrupoles with stacking error (standard deviation) higher than 5% and subsequently by applying the reciprocal test. The latter has been applied with a threshold value relatively high, 20%, in order to avoid losing too many measurement points, have a large inversion dataset that homogeneously covered the investigated subsurface, and this way precisely detect the areas with negative resistivity variation related to the injected water flow. By applying this data filtering, the number of measurements in each dataset have been reduced from 2594 to 2112. We can better explain the data acquisition and filtering process in the revised manuscript. See lines 72-84, and 100-102 in the Revised Manuscript.

- L114ff: This is unclear to me: why should 'slight' changes be more unreliable than stronger changes? Inversion artefacts, measurement errors can easily cause strong anomalies, which would result in significant changes. I would rather delete this sentence and mention the sources of uncertainty more neutrally.

**Reply:** For the inversion process of the datasets we defined an expected data error relatively high of 20% based on the reciprocal test, which is considered a reliable approach (Binley, 2015). Therefore, we do not think that very low resistivity changes (<10%) in the inverted tomograms can be considered reliable as the strong anomalies (>20%) to highlights the flow of the injected water. Obviously, also these higher % variation can be linked to inversion artefacts, measurement errors, changes in contact resistances, etc. However, in our opinion, they are more likely to be related to the injected water flow than lower variations. See lines 108-110 in the new Revised Manuscript.

- L125f and L130 and 136f: be careful here: the edges of the tomogram are the zones with the least data coverage/sensitivity, therefore anomalies tend to be extrapolated to depth. I would not necessarily conclude from this tomogram, that the resistive layer is not continuous, it may just not be detectable at the edges of the tomogram.

**Reply:** As mentioned in the second general comment reply, we agree with the comment of the reviewer that the bottom and the edges of the tomogram are the least sensitive zones. Nevertheless, as previously highlighted, we acquired a very high number of measurement points (2594) and even after the applied filtering, the inversion of each dataset was performed with 2112 measurements, which homogeneously covered the investigated pseudo-section. Furthermore, the acquisition scheme that we applied (Dipole-dipole multi-skip) has been specifically designed to guarantee high resolution in the shallower part of the subsurface and, at the same time, a large penetration depth. Moreover, we already knew the limits of the frozen layer from the ERT surveys carried out in the summer of 2021 (see Fig.1B). One of these survey lines was practically parallel to the one used for the infiltration experiment, but it was longer (48 channels with 3m spacing) and it allowed to define vertical and lateral boundaries of the discontinuous frozen layer along that direction. As suggested previously, we can insert a map of the rock glacier with the

location of the ERT survey lines performed in summer 2021 and the obtained resistivity section that better highlight the discontinuous structure of the frozen layer and helped us in the intepretation of the time-lapse experiment result. See revised Manuscript lines 56-58, 72-84, 124-131,152-154, and Figs.1 D-E in the Revised Manuscript.

- L134 and L163: unclear: what do you mean with 'upslope the permafrost body'? 'On top of'/'above' the permafrost body?

**Reply:** In this case, we meant that the injected water seems to flow in the South-West direction ("behind" the frozen body), not only down-slope in North-Est direction (maximum slope gradient). We can explain it better in the revised manuscript. See lines 124-131, and 141-143 in the Revised Manuscript.

- L171ff: again (even if I agree that one would expect the permafrost layer to act as an aquiclude), be careful to avoid over-interpretation: the resistive layer is very deep (relative to your measurement geometry) and in the least sensitive zone of your tomogram. Accordingly, resistivity changes would be the least obvious in this zone

**Reply:** As mentioned previously, we totally agree with the reviewer regarding the large uncertainties at the corners and at the bottom of the section. In the second general comment reply we already tried to explain our point of view and the reason why we are confident of our results. Nevertheless, we think that it will be useful for the manuscript to add these considerations about the sensitivity and the uncertainties of the results, but also that we tried to optimize the survey with a very high number of measurement points, the adopted acquisition scheme, and the position chosen to perform the infiltration test. See revised Manuscript lines 56-58, 72-84, 124-131,152-154, and Figs.1 D-E.

Fig. 1: can you indicate in B), which part of the rock glacier is shown in A)? Brown lines for ERT profiles are not well visible. Also consider to indicate the point of spring monitoring in front of the rock glacier.

**Reply**: We think that the reviewer is asking if we can indicate in B), which part of the rock glacier is shown in C) not in A). We can do it in the revised version and add a better plot of the brown lines in Fig.1B. In Fig. 1B and 1C, we can indicate also the position of the spring in front of the rock glacier. See Fig.1 in the Revised Manuscript.

- Fig.3: I would prefer to see the tomogram t0 plotted with a logarithmic colour scale (common standard in the community).

**Reply:** In the revised manuscript we will plot it in logarithmic scale. See Fig. 3 in the Revised Manuscript.

- as mentioned before I strongly recommend to plot the resistivity change based on logarithmic values, which will smooth the overall resistivity change and balance changes in low and high resistivity zones with a probable positive effect on interpretability…

**Reply:** In the revised manuscript we will plot the resistivity change based on logarithmic values, as discussed in the reply to the first general comment. See Fig. 3 in the Revised Manuscript and lines 104-108.

- indicate the point of injection in Fig. 3A (as far as I understand it is in the center of the profile) --> how do you explain resistivity decrease upslope of this point?? Snowmelt? Seasonal active layer thaw?

**Reply:** We thank the reviewer for the useful suggestion and in the revised manuscript we will plot the tomograms with an indication of the injection point. Regarding the negative % resistivity changes upslope the injection point, we presented the following explanation in lines 160-164: "…the presence of boulders and large rocks at various depths can lead to local reductions of the permeable area, causing funnel flow, and/or a splitting of flow paths (Hartmann et al., 2020). Splitting flow paths, due to the presence of large boulders, may have determined the infiltration of some injected water along the south-west direction and a local accumulation upslope of the injection point, which resulted in the observed negative resistivity variations." During the infiltration experiment there was no snow cover on the rock glacier and, since the infiltration experiment was carried out in few hours, we think that we can exclude any link with active layer thawing. Moreover, as specified in line 104, no rain or uncontrolled water contribution happened during the 24 hours of the experiment. See Fig.3 and 141-143 in the Revised Manuscript.